# The Physiological Basis of Alfalfa Plant Height Establishment

**DOI:** 10.3390/plants13050679

**Published:** 2024-02-28

**Authors:** Fang Jing, Shangli Shi, Wenjuan Kang, Jian Guan, Baofu Lu, Bei Wu, Wenjuan Wang

**Affiliations:** Key Laboratory of Grassland Ecosystem of Ministry of Education, College of Pratacultural Science, Gansu Agricultural University, Lanzhou 730070, China; jingf@st.gsau.edu.cn (F.J.); guanj@st.gsau.edu.cn (J.G.); lubf@st.gsau.edu.cn (B.L.); wub@st.gsau.edu.cn (B.W.); wangwenj@st.gsau.edu.cn (W.W.)

**Keywords:** plant height traits, leaf characteristics, photosynthetic physiology, endogenous hormones, reproductive period

## Abstract

Plant height plays an important role in crop yield, product quality, and cultivation management. However, the physiological mechanisms that regulate the establishment of plant height in alfalfa plants remain unclear. Herein, we measured plant height traits, leaf characteristics, photosynthetic physiology, cell wall composition, and endogenous hormone contents of tall- and short-stalked alfalfa materials at different reproductive periods. We analyzed the physiology responsible for differences in plant height. The results demonstrated that the number of internodes in tall- and short-stalked alfalfa materials tended to converge with the advancement of the fertility period. Meanwhile, the average internode length (IL) of tall-stalked materials was significantly higher than that of short-stalked materials at different fertility periods, with internode length identified as the main trait determining the differences in alfalfa plant height. Leaf characteristics, which are closely related to photosynthetic capacity, are crucial energy sources supporting the expression of plant height traits, and we found that an increase in the number of leaves contributed to a proportional increase in plant height. Additionally, a significant positive correlation was observed between plant height and leaf dry weight per plant during the branching and early flowering stages of alfalfa. The leaves of alfalfa affect plant height through photosynthesis, with the budding stage identified as the key period for efficient light energy utilization. Plant height at the budding stage showed a significant positive correlation with soluble sugar (SS) content and a significant negative correlation with intercellular CO_2_ concentration. Moreover, we found that alfalfa plant height was significantly correlated with the contents of indole-3-acetic acid in stem tips (SIAA), gibberellin A_3_ in leaves (LGA_3_), zeatin in stem tips (SZT), and abscisic acid in leaves (LABA). Further investigation revealed that SS, SIAA, and LGA_3_ contents were important physiological indicators affecting alfalfa plant height. This study provides a theoretical basis for understanding the formation of alfalfa plant height traits and for genetic improvement studies.

## 1. Introduction

Plant height serves as a comprehensive indicator of plant morphological structure and physiological function. It is a key determinant for plants’ absorption of light energy, photosynthesis, and nutrient partitioning, thus playing a crucial role in crop yield, product quality, and cultivation management [1]. In higher plants, the establishment of plant height significantly influences light distribution and penetration within the crop canopy, thereby impacting plant growth and biomass allocation [2]. The determination of plant height involves various developmental factors, with cell division and expansion of the apical and intermediate meristematic tissues playing a decisive role in stem elongation. The meristem of the stem tip continuously generates new internode tissue, which rapidly elongates through a balance of cell division and growth. Subsequently, it accumulates in the upper part of the internode to form secondary cell wall internodes that cease elongation [3]. Cell wall biosynthesis and amplification are crucial aspects influencing plant height. The cell wall determines plant height by regulating cell growth and internode elongation [4,5]. The cell wall forms a three-dimensional network comprising cellulose, hemicellulose, pectin, and proteins linked by covalent or noncovalent bonds [6]. Cellulose plays a significant role in regulating plant cell volume and determining cell size [7]. Cells attain their final size and shape following secondary cell wall development and initiate lignification after cell differentiation [8]. A suitable level of lignin deposition is necessary for plant development, as any reduction in lignin content can lead to plant dwarfing [9].

The stem serves as a vital nutrient organ positioned between the roots and leaves of plants, responsible for water and nutrient transportation [10]. Plant stem growth and development are intricately controlled by the recognition and response to both environmental and internal signals [11]. Environmental signals comprise light, temperature, water, and fertilizer. Among these factors, light is the primary environmental influence affecting plant growth and development, playing a pivotal role in photosynthesis and morphogenesis. Photosynthesis, contributing 90–95% of plant dry matter, is a crucial source for crop yield formation [12]. Stem elongation significantly determines a plant’s ability to compete for light. In low-light environments, plants increase internode length (IL) to elongate stems and access limited light resources [13]. Hormones are crucial internal signals in plants, and for several years, scientists have been studying the regulatory role of plant hormones. Phytohormones are small molecules that circulate in tissues or cells, traverse cell gaps and vascular bundles, and play important roles in plant morphogenesis, growth, and metabolism [14]. All physiological plant responses, including those influenced by hormones such as indole-3-acetic acid (IAA), cytokinin (CK), gibberellin (GA), and abscisic acid (ABA), occur at very low concentrations [15]. These hormones regulate cellular activities, including cell division, elongation, differentiation, and responses to biotic stress. IAA primarily promotes cell elongation during plant growth and development [16], whereas GA, a diterpenoid compound, mainly promotes cell expansion, differentiation, and proliferation [17]. CK is primarily involved in cell division and volume expansion, acting as a positive regulatory factor for stem meristem activity, thus playing a crucial role in stem development [18,19]. Plant hormones interact with each other, collectively regulating plant growth and development.

As the earliest cultivated and most widely distributed perennial high-quality leguminous forage worldwide, alfalfa (*Medicago sativa* L.) is known as the “King of Forage Grasses” because of its high grass yield, good quality, robust resistance, and broad adaptability [20,21]. Alfalfa cultivation in China is predominantly found in the northeast, north, and northwest regions. The cultivation of alfalfa has not only enhanced the ecological environment but also established alfalfa as a dominant grass species for improving both natural and artificial grasslands. The stem is the primary organ influencing plant height, aboveground biomass, and forage production, contributing to essential functions such as photosynthesis, nutrient storage, and plant regeneration [22]. During stem growth and development, the stem tip exhibits continuous growth, lateral branches emerge, and leaves are sequentially produced, collectively forming an extensive branching system [23]. Alfalfa, as a representative example, features the stem and meristem as the main biomass components. Its stem follows an “S” curve throughout the reproductive period, experiencing height increase. The highest growth rate occurs from the meristematic stage to the bud stage, peaking at the first flowering stage, following which growth nearly ceases [24]. Although existing studies on alfalfa growth mechanisms have mainly focused on factors such as drought, cold, and salinity resistance, few have explored the mechanisms underlying alfalfa plant height variations. Therefore, this study aimed to investigate the correlation between plant height traits and physiological indices of alfalfa at different reproductive periods by examining plant height-related traits, leaf characteristics, photosynthetic physiology, cell wall composition, and endogenous hormone contents during the establishment of plant height (branching, budding, and early flowering period) in both tall- and short-stalked alfalfa materials (Figure 1). Additionally, we analyzed the phenotypic traits and physiological basis contributing to differences in plant height during the height establishment period. The findings from this study are essential for laying a theoretical foundation for genetic improvements in plant height traits and the development of high-yielding varieties of alfalfa.

## 2. Results

### 2.1. Changes in Plant Height-Related Traits during Alfalfa Plant Height Establishment

As shown in Figure 2, the plant height of the tall- and short-stalked materials continued to grow from the branching stage to the first flowering stage, in which the plant height of the tall-stalked materials was significantly higher than that of the short-stalked materials at the budding and first flowering stages. There were certain differences in the growth rate of different alfalfa materials, but no change in pattern was observed. Notably, the growth rates of the two tall-stalked materials varied greatly, with WL525HQ having the largest growth rate at the branching stage and then decreasing with the advancement of the reproductive period, whereas Gannong No. 3 had a slower growth in the early stage before growing faster afterward. The plant height comprised the number of internodes (NI) and IL, in which the trends of plant height and NI were consistent. However, the trends of plant height and average IL with fertility period were opposite, and the average IL was the largest at the branching stage, declining gradually with the advancement of the fertility process. During growth and development, the NI in the tall- and short-stalked materials tended to be consistent with plant growth until the initial flowering stage, and the NI between the two materials differed insignificantly. The average IL of the tall-stalked materials during the entire growth period was significantly higher than that of the short-stalked materials (*p* < 0.05), indicating that IL was the main factor causing differences in plant height.

### 2.2. Changes in Leaf Characteristics during Alfalfa Plant Height Establishment

The leaf characteristic indices of alfalfa included the number of leaf blades per plant, leaf area (LA), leaf shape index, leaf dry weight per plant, and leaf–stem ratio (LSR) (Figure 3). The leaf shape index and leaf dry weight per plant were consistent with the growth trend of plant height, showing an increasing trend with the advancement of the reproductive period. The average IL was consistent with the growth trend of the LA, showing a decreasing trend with plant growth and development. During the alfalfa plant height establishment, the LA of the tall-stalked alfalfa materials was higher than those of the short-stalked materials, whereas the LSR was lower than that of the short-stalked materials. Furthermore, the dry weight of single plant leaves of the tall-stalked materials was significantly higher than that of short-stalked materials during the initial flowering stage. Among the tall-stalked materials, the dry weight of single plant leaves of Gannong No. 3 differed insignificantly from that of the short-stalked materials during the branching and budding stages but was significantly higher than that of short-stalked materials during the initial flowering stage, which may be related to the growth rate of Gannong No. 3.

### 2.3. Changes in Photosynthetic Parameters during Alfalfa Plant Height Establishment

As shown in Figure 4, the changes in photosynthetic parameters of the tall- and short-stalked alfalfa materials had a certain pattern, in which the transpiration rate (Tr), net photosynthetic rate (Pn), and stomatal conductance (Gs) tended to increase and then decrease with growth and development and reached a maximum at the budding stage. During plant height establishment, no significant difference in photosynthetic parameters was observed between the tall- and short-stalked materials, but the Tr and Pn values of the tall-stalked material Gannong No. 3 were outstanding. The intercellular CO_2_ (Ci) concentration of the short-stalked material WL343HQ was significantly higher than that of the other materials.

### 2.4. Changes in Photosynthetic Products during Alfalfa Plant Height Establishment

As shown in Figure 5, the soluble sugar (SS) content of the tall and short-stalked alfalfa materials differed significantly at different fertility periods, with the SS content of short-stalked materials decreasing with the advancement of the fertility period, whereas the trend of the tall-stalked materials was inconsistent. However, the SS content of the tall-stalked materials was higher than that of the short-stalked materials at the budding and early flowering stages. The sucrose (Suc) content differed significantly between the two short-stalked materials during the initial flowering stage. Furthermore, the Suc content of the tall-stalked materials was higher than that of the short-stalked materials at the budding stage. The starch (Sta) content of short-stalked materials varied significantly during the branching stage, but no significant difference was observed between the budding and early flowering stages. The Sta content of the tall-stalked materials differed significantly at the budding stage and was similar at the branching and first flowering stages.

### 2.5. Changes in Cell Wall Composition during Alfalfa Plant Height Establishment

As shown in Figure 6, the lignin content of the tall- and short-stalked alfalfa materials differed insignificantly during different fertility periods, whereas the fluctuation of cellulose and hemicellulose content was significant. In terms of cellulose content, the trend of changes in WL354HQ and Gannong No. 3 was consistent. There was a large difference in cellulose content between the short-stalked material at the branching and budding stages and an insignificant difference in cellulose content at the initial flowering stage, while there was an insignificant difference in cellulose content between the tall-stalked material at the branching and budding stages and a significant difference in content at the initial flowering stage. In terms of hemicellulose content, there was a large difference in hemicellulose content at the budding stage in the short-stalked material. There was a difference in the content at the branching stage in the tall-stalked material, while the difference at the budding stage and early flowering stage was small.

### 2.6. Changes in Endogenous Hormone Content during Alfalfa Plant Height Establishment

As shown in Figure 7, the fluctuation of the leaf zeatin (LZT) content of the short-stalked material at different reproductive periods was small and the trend was relatively consistent, whereas the fluctuation of the two tall-stalked materials was larger, among which the LZT content of Gannong No. 3 increased rapidly from the budding stage to the early flowering stage, and the LZT content was significantly higher than that of the other materials at the early flowering stage. In the stem tip, the variation trend of the stem tip zeatin (SZT) content in tall-stalked materials at different growth stages was inconsistent, whereas the SZT content of the two short-stalked materials showed an upward trend with the advancement of growth, and the SZT content of the short-stalked materials was significantly higher than that of the tall-stalked materials during the budding stage.

The leaf GA_3_ (LGA_3_) and stem tip GA_3_ (SGA_3_) contents of the alfalfa plants during the establishment stage showed opposite trends, with a higher LGA_3_ content at the branching stage than at the early flowering stage, whereas the SGA_3_ content was significantly higher at the early flowering stage than at the branching stage. Additionally, the LGA_3_ content of the tall-stalked materials during the budding stage was significantly higher than that of the short-stalked materials. However, there was a turning point in the early flowering stage, and the LGA_3_ content of the tall-stalked materials was lower than that of the short-stalked materials. Simultaneously, the SGA_3_ content of the tall-stalked materials was lower than that of the short-stalked materials during the budding stage but tended to be consistent during the initial flowering stage.

The variation in the trends of leaf IAA (LIAA) and stem tip IAA (SIAA) contents of the tall-stalked materials was relatively stable, with a decreasing trend in leaf blades and an increasing trend in stem tips as the reproductive period progressed. Among them, the LIAA and SIAA contents of the tall-stalked material WL525HQ were significantly higher than those of Gannong No. 3. The LIAA and SLIAA content of short-stalked materials fluctuated significantly at different reproductive periods, especially in WL354HQ. The LIAA content in WL354HQ rapidly decreases from the budding stage to the first flowering stage, while the SIAA content is significantly higher in the budding stage than in other stages.

The leaf ABA (LABA) and stem tip ABA (SABA) contents of alfalfa showed an overall increasing trend with the advancement of the reproductive period. The LABA content of the short-stalked material was significantly higher than that of the tall-stalked material. In the stem tip, the SABA contents of the two tall-stalked materials differed significantly, among which that of the tall-stalked material WL525HQ was significantly higher than that of the short-stalked material, whereas that of Gannong No. 3 differed insignificantly from that of the short-stalked materials during the branching and early flowering stages.

The leaf salicylic acid (LSA) contents of the two short-stalked alfalfa materials differed significantly at different reproductive periods, whereas those of the tall-stalked materials differed insignificantly. Moreover, the LSA contents of the tall-stalked materials were lower than those of the short-stalked materials at the budding and early flowering stages. In the stem tip, the difference was more significant in the tall-stalked materials at the branching stage, and the stem tip salicylic acid (SSA) content of the short-stalked materials differed insignificantly in different reproductive periods.

### 2.7. Association of Plant Height Traits with Leaf Characteristics and Physiological Indicators during Alfalfa Plant Height Establishment

The correlation analysis of plant height-related traits with leaf characteristics, photosynthetic physiological properties, cell wall components, and endogenous hormone contents of the tall- and short-stalked alfalfa materials at different reproductive periods showed the following results (Table 1). Plant height was significantly positively correlated with single plant leaf dry weight and SIAA content (*p* < 0.05), with correlation coefficients of 0.962 and 0.970, respectively. There was a significant negative correlation (*p* < 0.05) with LZT content, with a correlation coefficient of −0.903. The absolute values of the correlation coefficients were ranked in descending order as follows: single plant leaf dry weight > SIAA content > LZT content. The NI was significantly positively correlated with the SZT and SGA_3_ contents (*p* < 0.05). The average IL was significantly negatively correlated with LSR and LZT and LABA contents (*p* < 0.05).

At the budding stage, plant height was correlated significantly positively with mean IL (*p* < 0.01), with a correlation coefficient of 1.000; significantly and positively with SS content (*p* < 0.05), with a correlation coefficient of 0.971; and significantly and negatively with LSR, Ci, SZT, SGA_3_, and LABA (*p* < 0.05), with correlation coefficients of −0.934, −0.939, −0.974, −0.966, and −0.923, respectively. The absolute values of the correlation coefficients were ranked in descending order as follows: average IL > SZT content > SS content > SGA_3_ content > Ci concentration > LSR > LABA content. The NI was significantly positively correlated with the number of leaves per plant, LA, dry weight of leaves per plant, and SABA (*p* < 0.05). The average IL was significantly positively correlated with SS (*p* < 0.05), but negatively with LSR, Ci concentration, SZT, SGA_3_, and LABA (*p* < 0.05).

At the first flowering stage, plant height was highly significantly positively correlated with average IL (*p* < 0.01, correlation coefficient: 0.990), positively correlated with leaf dry weight (*p* < 0.05, correlation coefficient: 0.916), and negatively correlated with LGA_3_ and LABA contents (*p* < 0.05, correlation coefficients: −0.904 and −0.971, respectively). The correlation coefficients were ranked in the following order: average IL > LABA content > single leaf dry weight > LGA_3_ content. The NI was significantly positively correlated with Suc content (*p* < 0.05). The average IL was significantly negatively correlated with LABA content (*p* < 0.05).

### 2.8. Principal Component Analysis (PCA) of Height-Related Traits and Physiological Indicators during Alfalfa Plant Height Establishment

Through the correlation analysis of plant height-related traits, leaf characteristics, and physiological indices of alfalfa in different reproductive periods, the trait indices that were significantly correlated with alfalfa plant height were screened for PCA: the plant height-related traits (IL), leaf characteristics (number of leaves per plant [NLP], LA, LSR, and dry weight of leaves per plant [LDWP]), photosynthetic physiological indices (Ci concentration, SS content, and sucrose content), and hormone contents (LZT, SZT, LGA_3_, SGA_3_, LABA, SABA, and SIAA). In the PCA plots of the plant height-related traits, leaf characteristics, and physiological indices at different fertility periods (Figure 8a), the first and second principal components explained 48.5% and 18.9% of the total variables, respectively. The degree of dispersion of tall- and short-stalked materials during the budding stage was relatively large, which related closely to factors, including SS content, Suc content, and SZT. The degree of dispersion at the branching stage was the second largest, which related closely to factors, including IL, LA, LGA_3_, and LABA. The degree of dispersion at the first flowering stage was smaller, suggesting that the trait indicators differ significantly, thereby affecting plant height at different reproductive periods. In the PCA plot of the plant height-related traits, leaf characteristics, and physiological indices of the tall- and short-stalked alfalfa materials (Figure 8b), the first and second principal components explained 49.5% and 18.8% of the total variables, respectively. Among them, the degree of dispersion was greater in the short-stalked materials, which was affected by a combination of factors. The degree of dispersion was smaller in the tall-stalked materials, which related closely to SIAA, NLP, SABA, Suc content, SS content, and LGA_3_, indicating that these indicators were the main factors affecting alfalfa plant height.

### 2.9. Physiological Basis of Alfalfa Plant Height Establishment

To further clarify the physiological basis of alfalfa plant height establishment, this study screened out the important trait indices affecting alfalfa plant height: plant height traits (IL and NI), leaf characteristics (NLP, LDWP, LSR, and LA), light and physiological indicators (SS content, Suc content, and Ci concentration), and endogenous hormones (LGA_3_, SIAA, LABA, and SZT). A theoretical model of alfalfa plant height development at different growth stages was constructed (Figure 9). Overall, alfalfa plant height development was regulated by both environmental signals and endogenous hormones. Endogenous hormones regulate gene expression through signal transduction and thus affect alfalfa plant height. Correlation analysis showed that alfalfa plant height was significantly positively correlated with SIAA content at the branching stage (r = 0.970, *p* < 0.05) and significantly and negatively with LGA_3_ content at the early flowering stage (r = −0.904, *p* < 0.05). PCA showed that the tall-stalked alfalfa materials related closely to SIAA and LGA_3_ contents. It can be inferred that SIAA and LGA_3_ are important endogenous hormones that affect alfalfa plant height. IL is an important phenotypic trait indicator that determines alfalfa plant height. It was highly significantly positively correlated with plant height at the budding and early flowering stages (r = 1.000, r = 0.990, *p* < 0.01). Leaf characteristics, which are closely related to photosynthetic capacity, are an important source of energy that supports the expression of plant height-related traits. The tall-stalked alfalfa materials had more leaf numbers than the short ones, whereas the increase in leaf numbers increased alfalfa plant height to a certain extent. Furthermore, alfalfa plant height at the branching and first flowering stages was significantly positively correlated with LDWP (r = 0.962, r = 0.916, *p* < 0.05). The leaf characteristic indicators of alfalfa affect plant height through photosynthesis, and the budding stage is the key period for efficient utilization of light energy in alfalfa. The SS content of alfalfa at the budding stage was significantly positively correlated with plant height (r = 0.971, *p* < 0.05). PCA showed that SS content was the main photosynthetic physiological indicator causing differences in alfalfa plant height. In summary, it was shown that IL, LDWP, SS, SIAA, and LGA_3_ play an important determinant role in the establishment of alfalfa plant height.

## 3. Discussion

### 3.1. Phenotypic Indicators Causing Plant Height Differences during Alfalfa Plant Height Establishment

Alfalfa stem elongation is directly reflected by an increase in the NI and elongation of internodal tissues, ultimately manifesting as an increase in plant height [25]. Some researchers investigated the mechanism of stem elongation and found that IL and NI were significantly positively correlated with alfalfa plant height and that there was a trade-off between IL and NI, which showed some scale dependence [26]. For crops such as wheat, rice, and maize, the longer the IL, the higher the plant height [27,28]. Sun found that the NI in tall-stalked tomato P502 was the same as that of the wild-type, but the IL of each node was longer than that of the wild-type, and the cells in the internodes also showed significant growth [29]. The number of stem nodes of the soybean dwarf mutant differed insignificantly from that of wild-type Zp661, but the IL and cell length were shortened [30]. H032 semi-dominant dwarf plants showed a phenotype of reduced plant height and shortened IL of the stalks, but the number of nodes differed insignificantly [31]. In this study, we found that the NI in the tall- and short-stalked alfalfa materials tended to be consistent with the advancement of the growth period. The average IL of the tall-stalked materials at different growth stages was significantly higher than that of the short-stalked materials. Correlation analysis showed a highly significant and positive correlation between plant height and average IL during the budding and early flowering stages. These findings were consistent with those of previous reports; therefore, it can be predicted that IL is the main factor causing differences in plant height between tall and dwarf alfalfa.

As an important factor in alfalfa configuration, leaf characteristics include leaf number, LA, leaf dry weight, and LSR. Leaf characteristics are important factors that affect the canopy structure of the population and the forage yield and quality [32]. The increase in leaf number to a certain extent increases the effective photosynthetic area, which is conducive to the accumulation of dry matter. As an important leaf characteristic indicator that affects yield traits, leaf dry weight contributes 30–60% of alfalfa yield [33]. In this study, it was found that the number of leaves in the tall-stalked alfalfa materials was higher than that in the short-stalked materials, and the increase in leaf number to some extent increased the plant height of alfalfa. Furthermore, there was a significant positive correlation between plant height and single plant leaf dry weight during the branching and early flowering stages of alfalfa. In summary, the increase in plant height affected the number and weight of leaves to some extent. LSR is an important index for measuring forage quality and palatability because alfalfa leaves are rich in various nutrients. Therefore, the proportion of leaf dry weight determines the nutritional value of forage. When LSR is large, leaf-rich alfalfa has a high nutritive value, and the palatability is better [34,35]. In this study, plant height at the budding stage was significantly negatively correlated with LSR. One possible explanation is that the increase in plant height at the budding stage caused a greater increase in the stem than in leaves, which is consistent with the findings of Akin [36]. Alfalfa had the highest stem-to-leaf ratio at the nutrient stage, which gradually decreased from this budding stage to maturity, whereas plant height gradually increased.

### 3.2. Photosynthetic Physiological Indicators Causing Plant Height Differences during Alfalfa Plant Height Establishment

Photosynthetic parameters reflect the strength of photosynthesis to a certain extent, of which Pn is an important index for evaluating the photosynthetic capacity of a plant, Tr reflects the ability of a plant to regulate water balance, and higher Tr and Gs levels are favorable for the exchange of gasses between the outside world and the plant body [37]. During plant growth and development, light and parameter indicators are dynamic. Guo Zhili found that the Pn of millet at the booting stage was 247.7% and 32% higher than that at the jointing and filling stages, respectively, and the Tr was 100.4% and 104.7% higher, respectively [38]. Liu Yuxiu showed that Pn and Ci concentrations of control and super-tall wheat showed a trend of increasing and then decreasing from the tasseling stage to the grouting stage [39]. Zhao Changjiang found that during the reproductive growth stage of soybean, Pn and Gs showed a decreasing trend with the advancement in the fertility period, Tr first increased and then decreased, and Ci concentration showed an increasing trend [40]. In this study, it was also found that Tr, Pn, and Gs of the tall- and short-stalked alfalfa materials showed a trend of increasing and then decreasing with plant growth and development and reached a maximum at the budding stage. Therefore, it can be hypothesized that the budding stage is a critical period for the efficient use of light energy in alfalfa. This is consistent with Wu’s findings that the photosynthetic efficiency of alfalfa gradually increased with time starting from the regreening stage and began to decrease after the budding stage [41].

Photosynthetic products are an important material basis for plant growth and development, and the accumulation of photosynthetic products can directly affect the establishment of plant morphology [42]. In the nutrient tissues of plants, SS and starch are the main forms of carbohydrate storage, and sucrose is the main constituent component of soluble carbohydrates, which is involved in processes such as physiological metabolism and osmoregulation with starch [43,44]. The level of plant growth is directly related to the photosynthetic products, and during the rapid growth of plants, the consumption of SS and starch is greater, and higher photosynthetic products are favorable to plant growth and development, which significantly promotes plant height [45]. In this study, we also found that the SS content of taller culm materials was higher than that of shorter culm materials at the budding and first flowering stages, and the starch content of taller culm materials was higher than that of shorter culm materials at the budding stage. PCA showed that the SS content was an important factor influencing the differences in alfalfa plant height. The stem of a plant determines the vertical distribution of leaves, and the stem, as a “reservoir” for nutrient absorption and assimilation, largely determines the ability of the plant to photosynthesize. Tall culm materials have relatively long stem nodes and a uniform vertical distribution of leaf blades, and the upper and middle leaf blades receive sufficient light, thus accumulating a higher amount of photosynthetically active products than short culm materials, consistent with the results of a previous study [46]. At the individual and population levels, differences in the local light competition environment led to stronger negative feedback between the inputs to plant height and the consumption of photosynthetic products by small individuals within the cluster, i.e., asymmetric competition for light resources, and higher photosynthetic products favored plant growth and development, which significantly promoted plant height [47]. This study also found that alfalfa plant height at the budding stage was closely related to the contents of SS and sucrose, indicating that the budding stage was a critical period for the accumulation of photosynthetic products in alfalfa.

### 3.3. Endogenous Hormone Levels Causing Plant Height Differences during Alfalfa Plant Height Establishment

Plant hormones regulate plant growth and development by controlling the fundamental processes of cell division, expansion, and differentiation. Previous studies have shown that phytohormones such as IAAs, CKs, GAs, and ethylene play important roles in the regulation of plant height and the formation of plant structures [3]. IAA plays an important role throughout plant growth and development by regulating cell division, elongation, and differentiation [48]. When IAA synthesis is mutated in plants, it can directly promote stem elongation, indirectly promoting plant height increase by external application of IAA [49]. Chen Yujie showed that IAA plays a role in regulating the elongation of plant stem internode tissues, and the IL of tall castors receives more growth hormone regulation, which is manifested as longer cells and ILs [50]. In this study, we also found that the plant height of alfalfa at the branching stage was significantly positively correlated with the SIAA content. PCA indicated that SIAA was a main factor causing differences in the plant height of alfalfa.

CKs are widely involved in plant physiological and metabolic activities, mainly in the form of cis zeatin, trans zeatin, and dihydro zeatin, which are key hormones indispensable in plants [51,52]. This study showed a close relationship between tall- and short-stalked alfalfa materials and SZT at the budding stage, a positive correlation between plant height and SZT content in alfalfa at the branching stage, and a significant negative correlation during the budding stage. This indicates that CKs play a crucial role in cell division and volume expansion of stems at the branching stage, whereas SZT content during the budding stage inhibits cell division.

Gas promotes plant height and internode growth mainly by stimulating cell differentiation and elongation [53]. They promote internode elongation in crops such as soybean (*Glycine max*), rapeseed (*Brassica campestris*), and sugarcane (*Saccharum officinarum*), as well as internode elongation in dwarf plants, to achieve normal plant height [54]. During plant growth and development, exogenous GAs can also affect plant height and internode growth by inducing changes in endogenous hormones [55]. The results showed a close relationship between the tall-stalked alfalfa materials and LGA at the branching stage. The height of alfalfa plants was positively correlated with the SGA_3_ and LGA_3_ content at the branching stage, whereas it was significantly negatively correlated with the budding stage. There was a significant negative correlation between plant height and LGA_3_ content of alfalfa at the initial flowering stage. These results indicate that GA_3_ at the branching stage promotes plant height increase and internode elongation through cell differentiation and elongation and that the increase in GA_3_ does not cause plant height growth after the plant reaches a certain height. Therefore, the budding stage is a critical period for plant cell division and differentiation, and alfalfa enters the reproductive growth stage from nutrient growth after the budding stage.

ABA is a class of hormones that inhibit the growth of plants or isolated organs and is more abundant in mature or senescent tissues and in organs or tissues close to dormancy. It usually negatively regulates the growth of plant height [56]. It has been shown that the content of endogenous ABA in semidwarf rice varieties is higher than that in tall varieties, and exogenous ABA can inhibit internode elongation in deep rice. Further studies have shown that ABA mainly exerts its inhibitory effect on internode elongation by downregulating the expression of GA synthesis genes and cell division and elongation-related genes in the internodes [57]. In this study, we also found that the tall-stalked alfalfa material was closely related to the SABA content. Additionally, alfalfa plant height at the budding and early flowering stages was significantly negatively correlated with LABA content.

In plants, the hormones of each plant do not act independently. Plant hormones can independently regulate internode elongation, and their interactions can directly or indirectly regulate their elongation [58]. For example, IAA induces internode elongation by regulating the production of active GA_3_ in peas, whereas GA_3_ regulates internode elongation by regulating IAA synthesis and transportation in Arabidopsis. IAA and GA_3_ ultimately control plant growth and development by regulating cell division, expansion, elongation, and differentiation [59].

## 4. Materials and Methods

### 4.1. Plant Materials

The test materials were two tall-stalked and two short-stalked alfalfa materials with consistent fertility and stable traits selected through 2-year field trials. The two short-stalked materials were *M. sativa* L. ’WL354HQ’ and ‘WL343HQ,’ and the two tall-stalked materials were *M. sativa* L. ‘WL525HQ’ and ‘Gannong No. 3′. The basic shape of each variety is shown in Table 2.

### 4.2. Growth Conditions and Treatments

The experiment was conducted in August 2021 at the experimental field of Gansu Agricultural University, College of Grass Industry (34°05′ N, 105°41′ E, 1525-m altitude), Lanzhou City, Gansu Province, Northwest China. Four varieties per plot, each placed in three pots, were arranged in randomized blocks with three replications. Tillage soil was mixed with nutrient soil at a 5:1 mass ratio and packed into plastic pots with dimensions of 24 cm × 24 cm (outer diameter × depth). Then, they were planted in the field environment, with 2/3 of the pots buried in the soil, each repetition spaced 80 cm apart, and the pots spaced 10 cm apart. Alfalfa seeds that were fully grown and of uniform size were selected, sterilized with 10% sodium hypochlorite solution, and uniformly sown into the planted pots (30 seeds/pot). They were watered regularly to ensure normal plant growth. From June to August 2022, during the alfalfa plant height establishment period (branching, budding, and first flowering stages), randomly selected plants with consistent plant size and neat flowering were used to measure various indices.

### 4.3. Measurement Indices

#### 4.3.1. Measurement of Height-Related Traits

Plant height was measured from the base to the top of the alfalfa stem using a vernier caliper. The rate of increase in plant height was calculated as follows:Kw (cm·d^−1^) = (H_n+1_ − H_n_)/(T_n+1_ − T_n_)(1)
where H_n+1_ − H_n_ indicates the increase in plant height during two consecutive sampling attempts and T_n+1_ − T_n_ indicates the interval between two consecutive sampling attempts. The average IL was calculated after counting the NI. The NI was counted as that present in the main stem of alfalfa.
The average internode length = the height of the main stem/number of internodes.(2)

#### 4.3.2. Measurement of Leaf Characteristic Indicators

The third leaf blade of alfalfa plants from the flag leaf downward was selected, and the LA was determined using an LA meter [60]. The leaf shape index was calculated.
The leaf type index (LI) = leaf length/leaf width.(3)

Selected plants with uniform growth in the field were mowed and brought back to the laboratory to count NLP. After separating the stems and leaves, drying them to a constant weight at 80 °C, and determining LDWP and the dry weight of stems per plant, the LSR was calculated.

#### 4.3.3. Measurement of Photosynthetic Parameters

The third fully expanded leaflet from the flag leaf downward was selected to measure the photosynthetic parameters of alfalfa using a German portable photosynthesizer GFS-3000. The measurement was performed from 9:00 a.m. to 11:00 a.m., with sufficient illumination, a CO_2_ concentration of 400 µmol·m^−2^·s^−1^, and light intensity of 1200 µmol·m^−2^·s^−1^. The measurement indicators included Tr (umol·m^−2^·s^−1^), Pn (mmol·m^−2^·s^−1^), Ci concentration (mmol·mol^−1^), and Gs (mmol·m^−2^·s^−1^), which were measured in three replications with three leaves selected from each pot [61].

#### 4.3.4. Determination of Photosynthetic Products

The top downward third to fifth leaves of uniformly growing single plants were randomly selected for determination of the content of photosynthetic products. SS content was measured using the anthrone colorimetric method [62], sucrose content was measured using a reagent kit purchased from Solarbio Science & Technology (Beijing, China), and starch content was measured using the perchloric acid hydrolysis anthrone colorimetric method [63].

#### 4.3.5. Determination of Cell Wall Composition

For each material, a mix of plants with relatively uniform growth was selected according to the arrangement. After removing the leaves, the main stem branches, and the upper and lower two internodes, the leaves were dried to a constant weight at 80 °C, crushed, and passed through a stand-by 40-mesh sieve. Cellulose, hemicellulose, and lignin contents were determined using anthrone colorimetry [64], the dinitrosalicylic acid method [65], and the acetobromine method [66], respectively.

#### 4.3.6. Determination of Endogenous Hormone Content

Alfalfa plants with uniform growth in the field were selected to collect the stem tip and leaf blade. The stem tip was the part with the leaf removed, and the leaf blade was the third to fifth leaf blade down from the top. Each material from each plot was mixed, harvested, and divided into three equal parts for the determination of endogenous hormone content, and hormone extracts were prepared and analyzed as described by Jingfang [67]. The contents of endogenous hormones, namely LIAA, SIAA, LGA_3_, SGA_3_, LABA, SABA, LZT, SZT, LSA, and SSA, were determined via ultrafast liquid chromatography using a Waters Arc quadruple gradient.

### 4.4. Statistical Analyses

One-way analysis of variance followed by Tukey’s multiple range test were performed using SPSS 25.0 software at a 5% probability level. These data were organized and processed using Microsoft Excel 2010 software. Graphs were created using GraphPad 8.0.2 software, and PCA plots were generated using Originpro software. Graphs depicting the physiological basis of plant height establishment were drawn using Microsoft PowerPoint 2010.

## 5. Conclusions

The average IL of tall-stalked materials significantly exceeded that of short-stalked materials in various reproductive periods, highlighting IL as the primary factor influencing alfalfa plant height. Alfalfa plant height demonstrated a close association with NLP, with a significant positive correlation between plant height at the branching and first flowering stages and the dry weight of leaves per plant. Consequently, the increase in plant height not only affected the number of leaves but also increased their weight to some extent.

Tr, Pn, and Gs of alfalfa exhibited an initial tendency to increase followed by a decrease as growth progressed, reaching their peak at the budding stage. This suggests that the budding stage is a pivotal period for the efficient utilization of light energy in alfalfa. The accumulation and distribution of photosynthetic products within the plant body are fundamental for ensuring normal plant growth and development. Notably, SS content emerged as a major factor causing variations in alfalfa plant height.

The budding stage emerged as a critical period for cell division and differentiation in alfalfa. Subsequent experiments revealed that SIAA and LGA_3_ were closely correlated with alfalfa plant height, serving as important factors contributing to differences in plant height. This present study comprehensively analyzed external and internal factors influencing plant height development in tall-stalked alfalfa, considering plant height-related traits, leaf characteristics, photosynthetic physiology, and hormone content.

## Figures and Tables

**Figure 1 plants-13-00679-f001:**
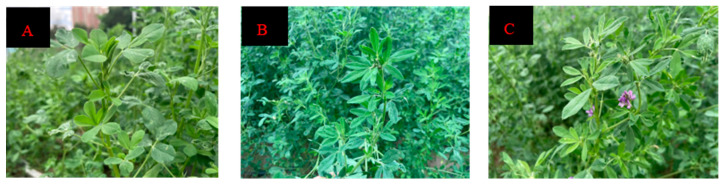
Alfalfa at different growth stages. (**A**) Branching period. (**B**) Budding period. (**C**) Early flowering period.

**Figure 2 plants-13-00679-f002:**
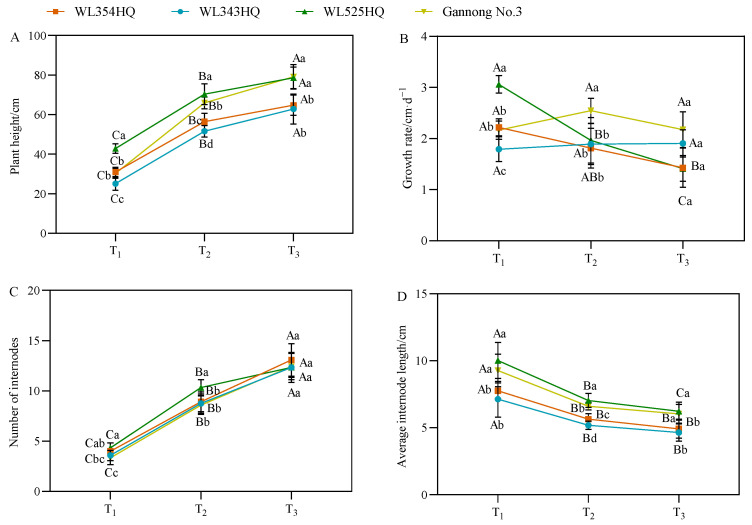
Indicators of plant height-related traits in tall- and short-stalked alfalfa materials. (**A**) Plant height. (**B**) Growth rate. (**C**) Number of internodes. (**D**) Average internode length. Different uppercase letters (A,B,C) indicate significant differences at different growth stages within the same variety (*p* < 0.05), whereas different lowercase letters (a,b,c,d) indicate significant differences among different varieties at the same growth stage (*p* < 0.05). Two short-stalked materials are WL354HQ and WL343HQ, and two tall-stalked materials are WL525HQ and Gannong No. 3. Orange, blue, green, and yellow represent the WL354HQ, WL343HQ, WL525HQ, and Gannong No. 3 varieties, respectively. T_1_, T_2_, and T_3_ represent the branching stage, the budding stage, and the first flowering stage, respectively.

**Figure 3 plants-13-00679-f003:**
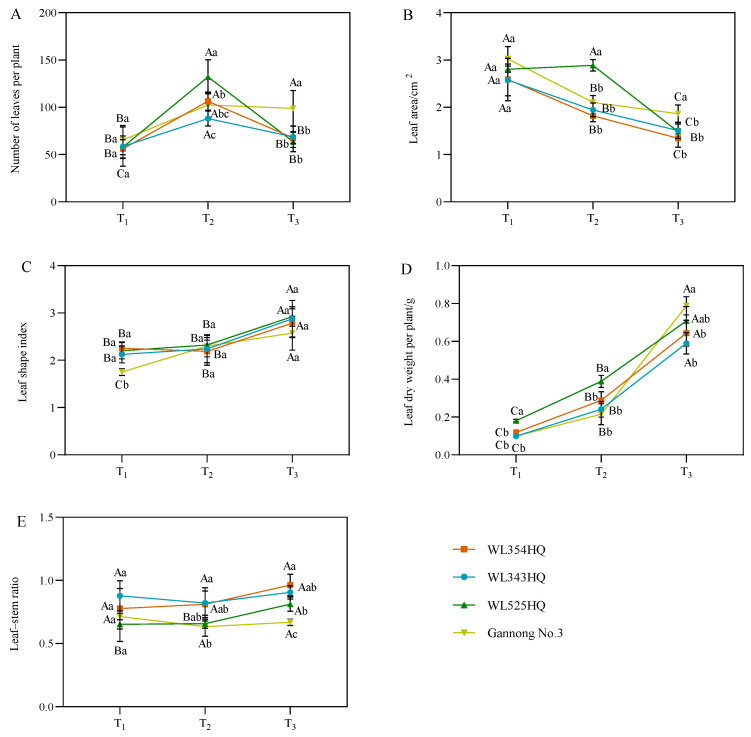
Leaf characterization indices of tall- and short-stalked alfalfa materials. (**A**) Number of leaves per plant. (**B**) Leaf area. (**C**) Leaf shape index. (**D**) Leaf dry weight per plant. (**E**) Leaf–stem ratio. Different uppercase letters (A,B,C) indicate significant differences at different growth stages within the same variety (*p* < 0.05), whereas different lowercase letters (a,b,c) indicate significant differences among different varieties at the same growth stage (*p* < 0.05). Two short-stalked materials are WL354HQ and WL343HQ, and two tall-stalked materials are WL525HQ and Gannong No. 3. Orange, blue, green, and yellow represent the WL354HQ, WL343HQ, WL525HQ, and Gannong No. 3 varieties, respectively. T_1_, T_2_, and T_3_ represent the branching stage, the budding stage, and the first flowering stage, respectively.

**Figure 4 plants-13-00679-f004:**
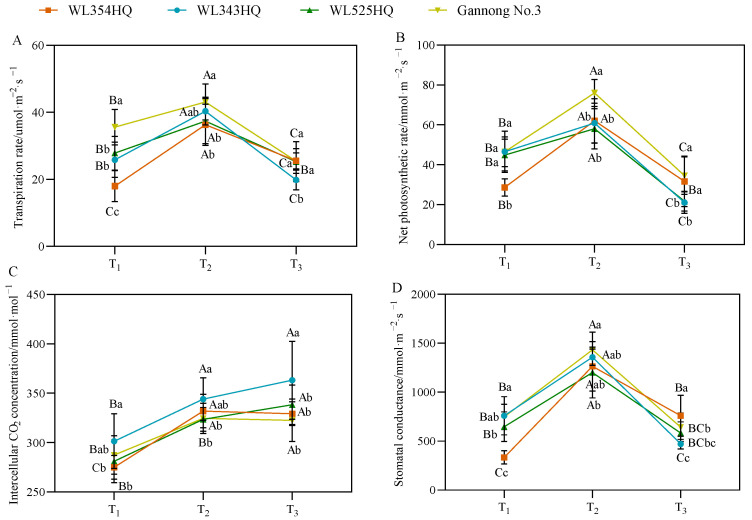
Photosynthetic parameters of tall- and short-stalked alfalfa materials. (**A**) Transpiration rate. (**B**) Net photosynthetic rate. (**C**) Intercellular CO_2_ concentration. (**D**) Stomatal conductance. Different uppercase letters (A,B,C) indicate significant differences at different growth stages within the same variety (*p* < 0.05), whereas different lowercase letters (a,b,c) indicate significant differences among different varieties at the same growth stage (*p* < 0.05). Two short-stalked materials are WL354HQ and WL343HQ, and two tall-stalked materials are WL525HQ and Gannong No. 3. Orange, blue, green, and yellow represent the WL354HQ, WL343HQ, WL525HQ, and Gannong No. 3 varieties, respectively. T_1_, T_2_, and T_3_ represent the branching stage, the budding stage, and the first flowering stage, respectively.

**Figure 5 plants-13-00679-f005:**
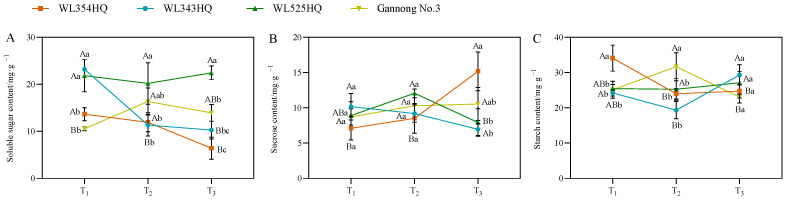
Photosynthetic product content of tall- and short-stalked alfalfa materials. (**A**) Soluble sugar content. (**B**) Sucrose content. (**C**) Starch content. Different uppercase letters (A,B,C) indicate significant differences at different growth stages within the same variety (*p* < 0.05), whereas different lowercase letters (a,b,c) indicate significant differences among different varieties at the same growth stage (*p* < 0.05). Two short-stalked materials are WL354HQ and WL343HQ, and two tall-stalked materials are WL525HQ and Gannong No. 3. Orange, blue, green, and yellow represent the WL354HQ, WL343HQ, WL525HQ, and Gannong No. 3 varieties, respectively. T_1_, T_2_, and T_3_ represent the branching stage, the budding stage, and the first flowering stage, respectively.

**Figure 6 plants-13-00679-f006:**
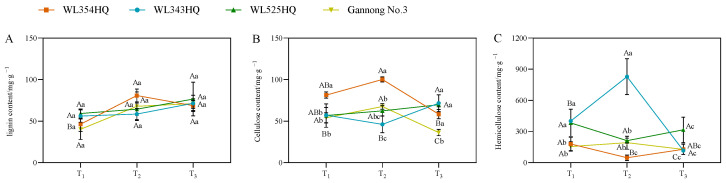
Cell wall composition of tall- and short-stalked alfalfa materials. (**A**) Lignin content. (**B**) Cellulose content. (**C**) Hemicellulose content. Different uppercase letters (A,B,C) indicate significant differences at different growth stages within the same variety (*p* < 0.05), whereas different lowercase letters (a,b,c) indicate significant differences among different varieties at the same growth stage (*p* < 0.05). Two short-stalked materials are WL354HQ and WL343HQ, and two tall-stalked materials are WL525HQ and Gannong No. 3. Orange, blue, green, and yellow represent the WL354HQ, WL343HQ, WL525HQ, and Gannong No. 3 varieties, respectively. T_1_, T_2_, and T_3_ represent the branching stage, the budding stage, and the first flowering stage, respectively.

**Figure 7 plants-13-00679-f007:**
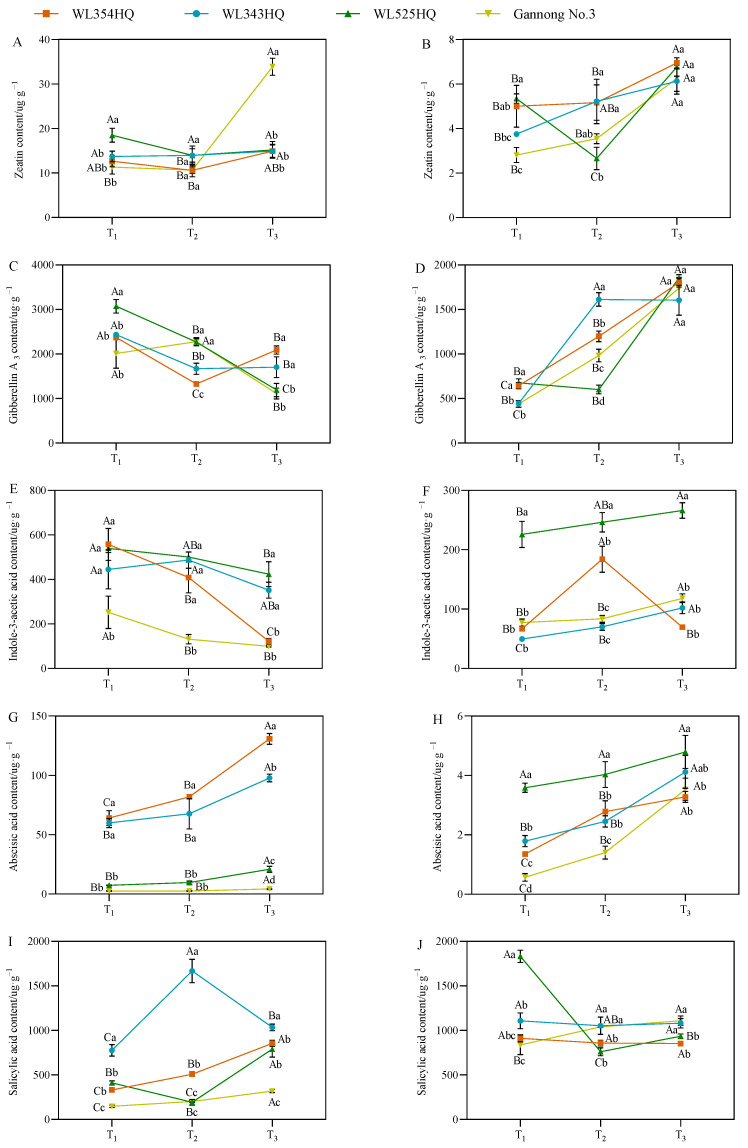
Endogenous hormone content in tall- and short-stalked alfalfa materials. Zeatin content in leaves (**A**) and stem tips (**B**). Gibberellin A_3_ content in leaves (**C**) and stem tips (**D**). Indole-3-acetic acid content in leaves (**E**) and stem tips (**F**). Abscisic acid content in leaves (**G**) and stem tips (**H**). Salicylic acid content in leaves (**I**) and stem tips (**J**). Different uppercase letters (A,B,C) indicate significant differences at different growth stages within the same variety (*p* < 0.05), whereas different lowercase letters (a,b,c,d) indicate significant differences among different varieties at the same growth stage (*p* < 0.05). Two short-stalked materials are WL354HQ and WL343HQ, and two tall-stalked materials are WL525HQ and Gannong No. 3. Orange, blue, green, and yellow represent the WL354HQ, WL343HQ, WL525HQ, and Gannong No. 3 varieties, respectively. T_1_, T_2_, and T_3_ represent the branching stage, the budding stage, and the first flowering stage, respectively.

**Figure 8 plants-13-00679-f008:**
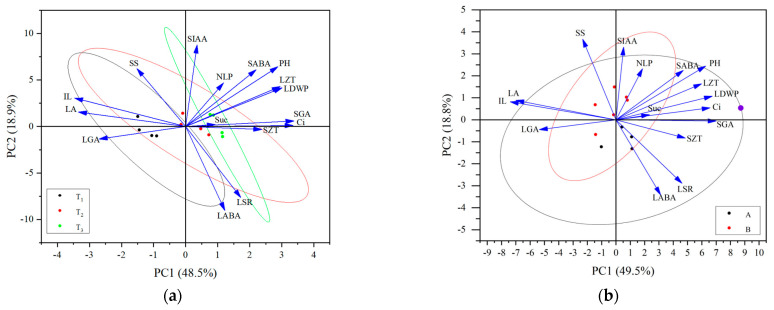
Principal component analysis of alfalfa plant height-related traits, leaf characteristics, and physiological indices. (**a**) The same shapes in black, red, and green represent different growth stages; (**b**) the same shapes in black and red represent short- and tall-stalked materials, respectively. Plant height (PH); average internode length (IL); number of leaves per plant (NLP); leaf area (LA); leaf dry weight per plant (LDWP); leaf–stem ratio (LSR); intercellular CO_2_ concentration (Ci); soluble sugar (SS); sucrose (Suc); indole-3-acetic acid in stem tips (SIAA); gibberellin A in leaves (LGA); gibberellin A in stem tips (SGA); abscisic acid in leaves (LABA); abscisic acid in stem tips (SABA); zeatin in leaves (LZT); zeatin in stem tips (SZT).

**Figure 9 plants-13-00679-f009:**
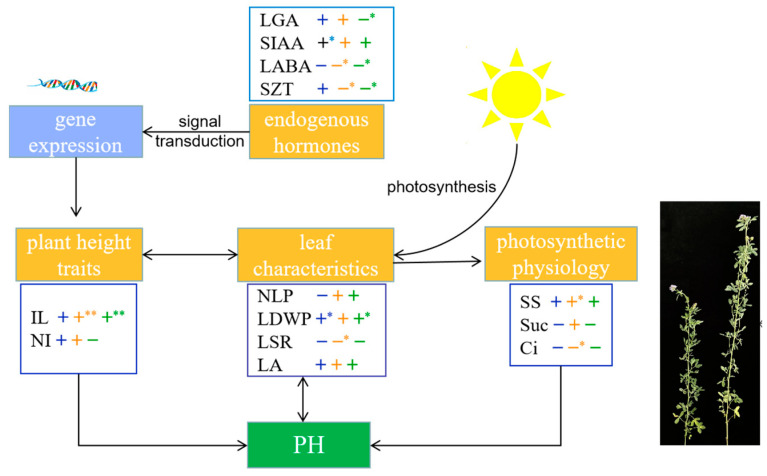
Physiological basis of alfalfa plant height establishment, with black arrows indicating interactions. Correlations (+ or −) between blue, orange, and green represent branching, budding, and early flowering stages, respectively. (**) Indicates a highly significant correlation at the 0.01 level, whereas (*) indicates a significant correlation at the 0.05 level.

**Table 1 plants-13-00679-t001:** Correlation analysis of alfalfa plant height-related traits with leaf characteristics and physiological indices.

Correlation		Branching Stage			Budding Stage			Initial Flowering Period		
		PH	NI	IL	PH	NI	IL	PH	NI	IL
NI		0.790	-	0.336	0.633	-	0.631	−0.450	-	−0.408
IL		0.843	0.336	-	1.000 **	0.631	-	0.990 **	−0.408	-
NLP		−0.207	−0.763	0.351	0.831	0.907 *	0.827	0.482	−0.189	0.363
LA		0.312	−0.332	0.773	0.818	0.914 *	0.819	0.633	−0.528	0.520
LI		0.272	0.804	−0.288	0.823	0.521	0.828	−0.352	−0.129	−0.246
LSR		−0.884	−0.452	−0.969 *	−0.934 *	−0.363	−0.936 *	−0.862	0.574	−0.784
LDWP		0.962 *	0.895	0.686	0.518	0.971*	0.515	0.916 *	−0.218	0.871
SS		0.168	0.434	−0.153	0.971 *	0.757	0.972 *	0.795	−0.681	0.837
Suc		−0.222	−0.318	−0.097	0.880	0.788	0.882	−0.198	0.949 *	−0.180
Sta		−0.016	0.335	−0.287	0.710	−0.081	0.710	−0.484	−0.414	−0.441
Tr		0.068	−0.532	0.559	0.061	−0.556	0.068	0.631	0.406	0.668
Pn		0.025	−0.376	0.339	0.200	−0.631	0.202	0.219	0.528	0.157
Ci		−0.598	−0.588	−0.431	−0.939 *	−0.467	−0.937 *	−0.587	−0.420	−0.579
Gs		−0.135	−0.500	0.203	−0.249	−0.841	−0.244	0.041	0.870	0.070
Lig		0.383	0.631	0.008	−0.011	−0.149	−0.020	0.600	−0.699	0.659
Cel		−0.102	0.339	−0.427	0.007	−0.094	−0.001	−0.474	−0.132	−0.381
Hem		0.203	0.388	−0.060	−0.567	−0.259	−0.560	0.587	−0.328	0.684
ZT	Blade	−0.903 *	−0.474	−0.978 *	−0.010	0.567	−0.006	0.605	−0.236	0.495
	Stem tip	0.652	0.978 *	0.147	−0.974 *	−0.687	−0.976 *	0.028	0.683	0.143
GA_3_	Blade	0.770	0.893	0.384	0.836	0.394	0.841	−0.904 *	0.778	−0.865
	Stem tip	0.770	0.955 *	0.356	−0.966 *	−0.739	−0.964 *	0.553	0.329	0.644
IAA	Blade	0.395	0.876	−0.160	−0.277	0.567	−0.279	0.016	−0.607	0.086
	Stem tip	0.970 *	0.755	0.818	0.541	0.867	0.535	0.631	−0.416	0.720
ABA	Blade	−0.596	0.020	−0.928 *	−0.923 *	−0.391	−0.926 *	−0.971 *	0.639	−0.942 *
	Stem tip	0.762	0.847	0.410	0.287	0.916 *	0.284	0.330	−0.710	0.394
SA	Blade	−0.338	0.079	−0.614	−0.856	−0.392	−0.852	−0.772	0.131	−0.698
	Stem tip	0.823	0.778	0.563	−0.509	−0.857	−0.503	0.203	−0.728	0.082

Note: (**) Indicates a highly significant correlation at the 0.01 level, whereas (*) indicates a significant correlation at the 0.05 level.

**Table 2 plants-13-00679-t002:** Source and type of alfalfa varieties used in the experiment.

Variety	Varietal Origin	Main Characters
WL354HQ	Beijing Zhengdao Ecological Technology Co., Beijing, China.	High leaf number and nutritional value
WL343HQ	Beijing Zhengdao Ecological Technology Co., Beijing, China.	Multiple branches, leaves, and high crude protein content
WL525HQ	Beijing Zhengdao Ecological Technology Co., Beijing, China.	Tall plants, high yield, and high protein content
Gannong No. 3	Key Laboratory of Grass Ecosystem, Ministry of Education, Gansu Agricultural University, Lanzhou, China.	Tall plant, stout stalks, high grass yield, well-developed lateral branches, and good growth

## Data Availability

The datasets presented in this article are not readily available because the data are part of an ongoing study. Requests to access the datasets should be directed to the corresponding author upon a reasonable request.

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
