# Peer review of "The Physiological Basis of Alfalfa Plant Height Establishment"

_plants, 2024, doi:10.3390/plants13050679_

Round 1
Reviewer 1 Report
Comments and Suggestions for Authors
Line 8:Clarifying the physiological mechanism of alfalfa plant height development is helpful for 8 genetic improvement of alfalfa plant height traits". Please modify it.
Line 14:while the average internode length of tall-stalked materials was significantly higher than that of short-stalked materials at different fertility periods. Please write results internode length of tall-stalked.
Please write increasing or decreasing percentage of all your parameters in abstract .
Line31: please remove keywords that are already used in title.
Please short the introduction part.
Please write description of T1,T2 and T3 below the figure legend.
Line 517:Medicago sativa L. Please itallic.
Conclusion is too long. please make short.
Comments on the Quality of English Language
English language correction is needed.
Author Response
For research article
Response to Reviewer 1 Comments
|
||
1. Summary |
|
|
Thank you very much for your comments and professional advice. These opinions help to improve academic rigor of our manuscript. Based on your suggestion and request, we have made corrected modifications on the revised manuscript. Meanwhile, the manuscript had be reviewed and edited by language services of NES. We hope that our revised manuscript can be get your consideration. Furthermore, we would like to show the details as follows: |
||
2. Questions for General Evaluation |
Reviewer’s Evaluation |
Response and Revisions |
Does the introduction provide sufficient background and include all relevant references? |
Can be improved |
We have made some improvements. |
Are all the cited references relevant to the research? |
Yes |
|
Is the research design appropriate? |
Can be improved |
We have optimized the formulation of the study design. |
Are the methods adequately described? |
Can be improved |
The research methodology was re-described. |
Are the results clearly presented? |
Yes |
|
Are the conclusions supported by the results? |
Can be improved |
We have condensed and re-summarized conclusions. |
3. Point-by-point response to Comments and Suggestions for Authors |
||
Comments 1: Line 8: Clarifying the physiological mechanism of alfalfa plant height development is helpful for 8 genetic improvement of alfalfa plant height traits". Please modify it. |
||
Response 1: We agree with this comment. Therefore, we have modified it. [edited text:However, the physiological mechanisms that regulate the establishment of alfalfa plant height are not clear. – page 1, line 8.] [We have updated text in the manuscript] |
||
Comments 2: Line 14: while the average internode length of tall-stalked materials was significantly higher than that of short-stalked materials at different fertility periods. Please write results internode length of tall-stalked. |
||
Response 2: Thank you for pointing this out. The article describes general trends in growth and development of tall and dwarf alfalfa material during different reproductive periods, and specific data were not presented in the results. |
||
Comments 3: Please write increasing or decreasing percentage of all your parameters in abstract . |
||
Response 3: Thank you for pointing this out. The overall idea of the article is to describe the changing trends of plant height traits, leaf characteristics, photosynthetic physiology, cell wall composition and endogenous hormone contents of tall and dwarf alfalfa materials at different reproductive periods, and analysis of the main factors contributing to the variances. Therefore, the results of the study do not include a detailed description of the specific percentage increase or decrease, so I apologize for not being able to provide all the data. |
||
Comments 4: Line31: please remove keywords that are already used in title. |
||
Response 4: We agree with this comment. Therefore, we have added the appropriate keywords. [page 1, line 31.] “[We have updated text in the manuscript]” |
||
Comments 5: Please short the introduction part. |
||
Response 5: We thank the reviewer for pointing this issue out. We have shorted the introduction part. [We have updated text in the manuscript] |
||
Comments 6: Please write description of T1,T2 and T3 below the figure legend. |
||
Response 6: I'm very sorry, it's my carelessness. We have writed the description of T1,T2 and T3 below the figure legend. [We have updated text in the manuscript] |
||
Comments 7: Line 517:Medicago sativa L. Please itallic. |
||
Response 7: We agree and have replaced [ page 2, line 80.] [We have updated text in the manuscript] |
||
Comments 8: Conclusion is too long. please make short. |
||
Response 8: We agree and have shorted the conclusion. [We have updated text in the manuscript] |
||
4. Response to Comments on the Quality of English Language |
||
Point 1: English language correction is needed. |
||
Response 1: The manuscript had be reviewed and edited by language services of NES. |
||
5. Additional clarifications |
||
[We have checked the full text and revised the reviewers' suggestions] |

Reviewer 2 Report
Comments and Suggestions for Authors
Authors have used "construction" word with height which is looking odd, even if authors only write height and remove the world construction the meaning will remain same. So consider the suggestion.
Author Response
For review article
Response to Reviewer 2 Comments
|
||
1. Summary |
|
|
Thank you very much for your comments and professional advice. These opinions help to improve academic rigor of our manuscript. Based on your suggestion and request, we have made corrected modifications on the revised manuscript. Meanwhile, the manuscript had be reviewed and edited by language services of NES. We hope that our revised manuscript can be get your consideration. Furthermore, we would like to show the details as follows:
|
||
2. Questions for General Evaluation |
Reviewer’s Evaluation |
Response and Revisions |
Does the introduction provide sufficient background and include all relevant references? |
Yes |
|
Are all the cited references relevant to the research? |
Yes |
|
Is the research design appropriate? |
Yes |
|
Are the methods adequately described? |
Yes |
|
Are the results clearly presented? |
Yes |
|
3. Point-by-point response to Comments and Suggestions for Authors |
|
|
Comments 1: Authors have used "construction" word with height which is looking odd, even if authors only write height and remove the world construction the meaning will remain same. So consider the suggestion.
|
||
Response 1: Thank you for pointing this out. I think this may be caused by my improper use of words. The purpose of this paper is to describe the changes in phenotypic traits and physiological indicators of alfalfa in the process of plant height establishment, if only the use of plant height, it can not be reflected in the period of the process of this dynamic change. After carefully considering the reviewer's comments, we have replaced the "construction" to "establishment"。 [We have updated text in the manuscript] |
||
4. Additional clarifications |
||
[We have checked the full text and revised the reviewers' suggestions] |

Reviewer 3 Report
Comments and Suggestions for Authors
The authors investigated the physiological mechanism of alfalfa plant height development by measuring the plant height traits, leaf characteristics, photosynthetic activity, cell wall composition and endogenous hormone contents of tall (WL525HQ and Gannong No3) and dwarf (WL343HQ and WL354HQ) varieties alfalfa at different reproductive periods (branching stage, budding stage and initial flowering stage). The manuscript is well-written, the authors obtained interesting results that are properly discussed.
I have the following comments and suggestions:
Since the description of Materials and Methods is at the end of the manuscript, it will be useful for readers to mention which are tall and dwarf alfalfa varieties, as well as the description of T1-T3 in the legend of Figure 1
At some points, the description of the results obtained is too general:
Line 115 – “In addition, the average daily growth rates of taller materials at branching and bud stage were higher than those of shorter materials” - The results presented in Figure 1B showed that the growth rate of WL525HQ was higher at T1, while that of Gannong No3 was higher at T2.
Line 141 – “In the process of alfalfa plant height establishment, the number of leaves per plant and leaf area of taller alfalfa materials were higher than that of shorter stem materials overall…” – Leaf number of WL525HQ was significantly higher only at T2 and that of Gannong No3 – at T3. The results showed that the main differences in the number of leaves per plant and leaf area were observed between WL525HQ and WL343HQ at T2.
Line 183 – “The Sta content of tall 183 stem materials showed significant differences during the budding stage, and was higher than that of short stem materials” – This is true mainly for Gannong No3, while the starch content of WL525HQ was similar to that of WL354HQ but higher than WL343HQ.
Line 232 – “the SIAA content of tall materials was higher than that of dwarf materials during branching and early flowering stages”- This is true only for WL525HQ
The description of results presented in Figure 5 is very brief. It was written that “There was no obvious pattern of change in the cell wall composition of the tall and short-stalked alfalfa materials in different fertility periods”. However, significant changes in cellulose and hemicellulose content between the two dwarf alfalfa materials were evident at T1 and T2.
Author Response
For research article
Response to Reviewer 3 Comments
|
||
1. Summary |
|
|
Thank you very much for your comments and professional advice. These opinions help to improve academic rigor of our manuscript. Based on your suggestion and request, we have made corrected modifications on the revised manuscript. Meanwhile, the manuscript had be reviewed and edited by language services of NES. We hope that our revised manuscript can be get your consideration. Furthermore, we would like to show the details as follows: |
||
2. Questions for General Evaluation |
Reviewer’s Evaluation |
Response and Revisions |
Does the introduction provide sufficient background and include all relevant references? |
Yes |
|
Are all the cited references relevant to the research? |
Yes |
|
Is the research design appropriate? |
Yes |
|
Are the methods adequately described? |
Yes |
|
Are the results clearly presented? |
Can be improved |
We have recharacterized the problematic areas of the results. |
Are the conclusions supported by the results? |
Yes |
|
3. Point-by-point response to Comments and Suggestions for Authors |
||
Comments 1: Since the description of Materials and Methods is at the end of the manuscript, it will be useful for readers to mention which are tall and dwarf alfalfa varieties, as well as the description of T1-T3 in the legend of Figure 1 |
||
Response 1: I'm very sorry, it's my carelessness. We've added tall- and short-stalked alfalfa materials, as well as the description of T1-T3 in the legend. [We have updated text in the manuscript] |
||
Comments 2: Line 115 – “In addition, the average daily growth rates of taller materials at branching and bud stage were higher than those of shorter materials” - The results presented in Figure 1B showed that the growth rate of WL525HQ was higher at T1, while that of Gannong No3 was higher at T2. |
||
Response 2: Agree. We have deleted this part of the text. [edited text: In addition, the average daily growth rates of taller materials at branching and bud stage were higher than those of shorter materials] [We have updated text in the manuscript] |
||
Comments 3: Line 141 – “In the process of alfalfa plant height establishment, the number of leaves per plant and leaf area of taller alfalfa materials were higher than that of shorter stem materials overall…” – Leaf number of WL525HQ was significantly higher only at T2 and that of Gannong No3 – at T3. The results showed that the main differences in the number of leaves per plant and leaf area were observed between WL525HQ and WL343HQ at T2. |
||
Response 3: Thank you for pointing this out. We have modified this section. [edited text: During the alfalfa plant height establishment, the LA of the tall-stalked alfalfa materials were higher than those of the short-stalked materials, whereas the LSR was lower than that of the short-stalked materials. –page 4, line 143.] [We have updated text in the manuscript] |
||
Comments 4: Line 183 – “The Sta content of tall stem materials showed significant differences during the budding stage, and was higher than that of short stem materials” – This is true mainly for Gannong No3, while the starch content of WL525HQ was similar to that of WL354HQ but higher than WL343HQ. |
||
Response 4: We have redescribed this section. [edited text: The starch content of the tall-stalked materials differed significantly at the budding stage and was similar at the branching and first flowering stages. page 6, line 195.] [We have updated text in the manuscript] |
||
Comments 5: Line 232 – “the SIAA content of tall materials was higher than that of dwarf materials during branching and early flowering stages”- This is true only for WL525HQ |
||
Response 5: We agree and have modified. [edited text: The LIAA and SLIAA content of short-stalked materials fluctuated significantly at different reproductive periods, especially in WL354HQ. The LIAA content in WL354HQ rapidly decreases from the budding stage to the first flowering stage, while the SIAA content is significantly higher in the bud stage than in other stages. page 7, line 256.] [We have updated text in the manuscript] |
||
Comments 6: The description of results presented in Figure 5 is very brief. It was written that “There was no obvious pattern of change in the cell wall composition of the tall and short-stalked alfalfa materials in different fertility periods”. However, significant changes in cellulose and hemicellulose content between the two dwarf alfalfa materials were evident at T1 and T2. |
||
Response 6: Thank you for pointing this out. We have added a description of this section. [edited text: In terms of cellulose content, the trend of changes in WL354HQ and Gannong No.3 was consistent. There was a large difference in cellulose content between the short-stalked material at branching and budding stage, and an insignificant difference in cellulose content at initial flowering stage. While there was an insignificant difference in cellulose content between the tall-stalked material at branching and budding stage, and a significant difference in content at initial flowering stage. In terms of hemicellulose content, there was a large difference in hemicellulose content at the budding stage in the short-stalked material. There was a difference in the content at the branching stage in the tall-stalked material, while the difference at the budding stage and early flowering stage was small. page 6, line 211.] [We have updated text in the manuscript] |
||
4. Response to Comments on the Quality of English Language |
||
Point 1: English language correction is needed. |
||
Response 1: The manuscript had be reviewed and edited by language services of NES. |
||
5. Additional clarifications |
||
[We have checked the full text and revised the reviewers' suggestions] |
